# A Multianalyte Electrochemical Genosensor for the Detection of High-Risk HPV Genotypes in Oral and Cervical Cancers

**DOI:** 10.3390/bios12050290

**Published:** 2022-05-02

**Authors:** Thanyarat Chaibun, Patcharanin Thanasapburachot, Patutong Chatchawal, Lee Su Yin, Sirimanas Jiaranuchart, Patcharee Jearanaikoon, Chamras Promptmas, Waranun Buajeeb, Benchaporn Lertanantawong

**Affiliations:** 1Biosensors Laboratory, Department of Biomedical Engineering, Faculty of Engineering, Mahidol University, Nakhon Pathom 73170, Thailand; thanyarat.chhh@gmail.com (T.C.); chamras.pro@mahidol.ac.th (C.P.); 2Department of Oral Medicine and Periodontology, Faculty of Dentistry, Mahidol University, Bangkok 10400, Thailand; 3Research Office, Faculty of Dentistry, Mahidol University, Bangkok 10400, Thailand; patcharanin.tha@mahidol.edu; 4Center for Research and Development of Medical Diagnostic Laboratories, Faculty of Associated Medical Sciences, Khon Kaen University, Khon Kaen 40002, Thailand; patutong@gmail.com (P.C.); patjea@kku.ac.th (P.J.); 5Faculty of Applied Sciences, AIMST University, Bedong 08100, Malaysia; su_yin@aimst.edu.my; 6Centre of Excellence for Omics-Driven Computational Biodiscovery (ComBio), AIMST University, Bedong 08100, Malaysia; 7Dental Clinic, Chulabhorn Hospital, Chulabhorn Royal Academy and Department of Oral and Maxillofacial Surgery, Faculty of Dentistry, Chulalongkorn University, Bangkok 10330, Thailand; sirimanas.jia@pccms.ac.th

**Keywords:** oral cancer, oral squamous cell carcinoma, cervical cancer, human papilloma virus, electrochemical biosensors, cancer diagnosis, nanomaterials

## Abstract

Infection with high-risk human papillomavirus (HPV) is a major risk factor for oral and cervical cancers. Hence, we developed a multianalyte electrochemical DNA biosensor that could be used for both oral and cervical samples to detect the high-risk HPV genotypes 16 and 18. The assay involves the sandwich hybridization of the HPV target to the silica-redox dye reporter probe and capture probe, followed by electrochemical detection. The sensor was found to be highly specific and sensitive, with a detection limit of 22 fM for HPV-16 and 20 fM for HPV-18, between the range of 1 fM and 1 µM. Evaluation with oral and cervical samples showed that the biosensor result was consistent with the nested PCR/gel electrophoresis detection. The biosensor assay could be completed within 90 min. Due to its simplicity, rapidity, and high sensitivity, this biosensor could be used as an alternative method for HPV detection in clinical laboratories as well as for epidemiological studies.

## 1. Introduction

Oral cancer is one of the most common and is considered the sixth most widespread cancer worldwide [1]. More than 90% of all oral cancers are oral squamous cell carcinomas (OSCC). Human papillomavirus (HPV) infections are considered to be the one of the most critical risk factors for OSCC [2,3]. Similarly, for cervical cancers, HPV has also been well established as the main risk factor. HPV are non-enveloped, double-stranded DNA viruses from the Papillomaviridae family that infect the mucosal and cutaneous epithelia. The International Agency of Research on Cancer (IARC) has declared that HPV-16, followed by HPV-18, are among the most common ‘high-risk’ genotypes due to their oncogenic potential. These HPV genotypes have been linked to OSCC, oropharyngeal squamous cell carcinomas (OPSCC) [4,5,6,7,8,9,10], and cervical lesions.

Identification of HPV in oral cancers is important as the clinical outcome and prognosis may differ from HPV-negative cases [11]. Diagnosis is conducted by the physical assessment and microscopic examination of tissues or cells. A screening tool that could provide early detection of HPV presence in oral samples is crucial for appropriate diagnosis and prompt treatment. Genotyping to identify HPV-16 and -18 has been recommended as a screening tool for cervical cancer for patients who have normal cytology before going for colposcopy [12]. HPV genotyping could provide additional information for risk assessment, prognosis, and therapy such as the use of therapeutic vaccines that target specific HPV genotypes [10].

Current methods for HPV detection include cytology, histopathology, immunohistochemistry, and real-time polymerase chain reaction (qPCR) [7]. Commercially available HPV tests such as Hybrid Capture II and qPCR-based kits are now widely used in the clinical laboratory setting. The Hybrid Capture assay is based on hybridization, of which 13 high-risk and five low-risk HPV genotypes can be detected using automation [13]. The Hybrid Capture assay requires a high amount of DNA samples due to a lack of DNA amplification. PCR-based detection is now the gold standard for HPV genotyping. Different HPV genotypes can be identified using genotype-specific primers and different fluorescent label probes in a single reaction. However, qPCR assays are limited in its use in a low resource setting or for epidemiological approach.

Electrochemical DNA biosensors have been developed as a rapid, portable, economical, sensitive, and specific detection tool for various targets. Many DNA biosensors have been developed for HPV-16 and HPV-18 detection. Espinosa et al. [14] and Ramesh et al. [15] used impedance-based DNA biosensors to detect HPV-16. Pareek et al. [16] and Mahmoodi et al. [17] used gold nanoparticle-based DNA biosensors to detect HPV-16 and HPV-18, respectively. However, these studies and many others have mainly focused on the detection of a single HPV genotype in cervical cancer samples or using synthetic DNA targets.

Considering the common etiology of oral and cervical cancers, we developed an electrochemical DNA biosensor that could detect both HPV-16 and HPV-18 genotypes. Furthermore, we evaluated the biosensor with both cervical and oral samples, which have different sample matrices. Instead of relying on different diagnostic tests for oral and cervical cancers, this “2-in-1” assay presents a single test for both cancers.

Our biosensor platform uses silica nanoparticles loaded with two different redox indicators and magnetic bead-capture probes that specifically target the HPV E6/E7 oncogenes of HPV-16 and HPV-18. The analytical performance of the developed biosensor was compared with the results of nested PCR.

## 2. Materials and Methods

### 2.1. Chemicals, Kits, and Instrumentation

All buffer solutions were prepared with purified Milli-Q water (18.2 MΩ). Poly(allylamine hydrochloride), (PAA MW~50,000), poly(sodium 4-styrenesulfonate) (PSS, MW~70,000), tetraethyl orthosilicate (TEOS), acridine orange (AO), and avidin from egg white were purchased from Sigma-Aldrich (St. Louis, MO, USA). Sodium chloride, methylene blue (MB), 2-propanol, 25% ammonium solution, 28–30% ammonium hydroxide, acetone, and 95% ethanol were purchased from Merck (Darmstadt, Germany). Dynabeads^TM^ MyOne^TM^ Streptavidin T1 was purchased from Thermo Fisher Scientific (Waltham, MA, USA). AccuPower 2× Greenstar qPCR Master Mix was purchased from Bioneer (Daejeon, Korea), 2× PCR Master Mix Solution (i-StarTaq) was purchased from iNtRON Biotechnology (Seongnam, Korea).

For electrochemical measurements, screen-printed carbon electrodes (SPCE) were obtained from Quasense (Bangkok, Thailand) and the PalmSens4 potentiostat with PSTrace software version 5.8 was obtained from Palmsens (Houten, The Netherlands).

### 2.2. Design of Primers and Probes

Nested PCR primers for HPV-16 and HPV-18 were designed based on the E6/E7 HPV oncogenes as described in previous studies [18,19]. The probes were designed to bind to the PCR amplicons and tagged with a biotin moiety at either the 3′ or the 5′ end. The mismatched linear targets (mLT) contain one or three mismatched bases. The oligonucleotide sequences are listed in Table 1. PCR primers were purchased from Bionics (Hongcheon, Korea), while probes and DNA linear targets (LT) were purchased from Integrated DNA Technologies Pte. Ltd. (Singapore).

### 2.3. Preparation of the Silica Seed Nanoparticles

Silica seed nanoparticles (SiNPs) were synthesized and the redox active dye was incorporated into the SiNPs by the modified method described previously [20,21]. First, 1.375 mL of TEOS was added to 9.6 mL of 2-propanol under slow stirring. A mixture of 0.5 mL of 25% (*v/v*) ammonia and 1.025 mL of Milli-Q water were added with constant stirring for 1 h, and the solution was heated to 50 °C to produce silica seed particles. Silica seed solution was added to 5 mL of TEOS, 227.5 mL of 2-propanol, and 44.5 mL of 8.29% (*v/v*) ammonia solution. Then, 91.25 mL of TEOS was added to the mixture at a rate of 0.5 mL/min, and the reaction was allowed to continue for an additional 1 h with fast stirring under 50 °C. Finally, the silica microsphere particles were isolated by centrifugation at 10,000 rpm for 10 min. The pellet was washed with Milli-Q water and centrifuged at 8000 rpm for 5 min. The washing step was repeated four times. The silica pellet was obtained by drying at 105 °C in an oven.

### 2.4. Loading of SiNPs with Redox-Active Dye

First, 0.3 g of silica powder was added into 10.9 mL of 2-propanol containing 1.5 × 10^−^^5^ mol of a redox dye (MB for HPV-16 and AO for HPV-18) and 0.55 mL of TEOS were mixed by sonication until it formed a colloidal suspension. After that, a separate solution of 25% ammonia diluted with 12 mL of Milli-Q water was added into the mixture. The reaction was allowed to proceed for 2 h at 40 °C. The pellet was washed with Milli-Q water and centrifuged at 8000 rpm for 5 min. This step was repeated four times and dried at 105 °C in an oven. SiNPs and dye incorporated silica (Si-dye) were characterized by scanning electron microscope (SEM, JMS-6610LV by JEOL Ltd., Tokyo, Japan).

### 2.5. Preparation of the Reporter Probe

A total of 10 mg/mL silica redox-active dye (Si-dye) was sequentially incubated in 0.2 mL of PAA and PSS polyelectrolytes (10 mg/mL in 0.5 M NaCl) at room temperature for 30 min, respectively. After washing to remove the excess polyelectrolytes, the particles were dispersed in 1 mL of 10 mM phosphate buffer (PB, pH 7.0). Next, 10 µL of avidin (21.14 mg/mL) was added into the solution and incubated at 37 °C for 1 h 30 min. The avidin-coated Si-dye/PAA-PSS (referred to as Si-dye/Avidin) particles were resuspended in 2 mL of 0.1 M PB (pH 7.0). The conjugation of Si–Avidin with the reporter probe (RP) was achieved by mixing 0.1 mL of Si–Avidin with 1 μL of 10 μM RP and 99 μL of 10 μM blocking probe (BP). Both RP and BP contain a biotin moiety at the 3′ end, which can bind to avidin. HPV-16-RP was conjugated with Si-MB/Avidin and HPV-18-RP was conjugated with Si-AO/Avidin. The mixture was incubated at room temperature for 30 min. The RP-conjugated to Si-dye/Avidin (referred to as Si-RP) was resuspended with 250 μL of 1 M PB and kept at 4 °C until use.

### 2.6. Preparation of Capture Probe

Four µL of 100 µM capture probe (CP), 12 μL of 100 µM blocking probe, and 100 µL of Dynabeads^TM^ MyOne^TM^ Streptavidin T1 (MNB, 10 μg/μL) were mixed and allowed to incubate at room temperature for 30 min with gentle agitation. The CP-MNB were separated from the unbound CP by magnetic separation, followed by washing three times with 20 mM PB (pH 7.0). Finally, the CP-MNB was resuspended in 100 μL of 20 mM PB and stored at 4 °C until use.

### 2.7. Target Hybridization and Signal Detection

First, 18 μL of DNA target was mixed with 2 μL of CP-MNB and incubated at 50 °C for 30 min. The separation of DNA target bound to CP-MNB (referred to as CP-MNB-Target) from the unbound CP-MNB was performed by magnetic separation, followed by the washing step. Next, 20 μL of Si-RP was added to the CP-MNB-Target mixture and incubated at 50 °C for 30 min to allow the sandwich hybridization to occur. After that, the separation of bound and unbound targets was facilitated by magnetic separation. Finally, the pellet was re-suspended with 0.1 M phosphate buffer containing 0.1 M potassium chloride (PB/KCl) (pH 7.0). This solution was pipetted onto the SPCE. Differential pulse voltammetry (DPV) was performed by scanning from −0.5 to −0.1 V (for HPV-16) and 0.2 to 1.0 V (for HPV-18), with a step potential of 0.01 V, time equilibration of 150 s, modulation amplitude of 0.05 V with an interval time of 0.05 s, and 0.1 V/s scan rate.

### 2.8. Sensitivity and Specificity of the Assay

The sensitivity of the assay was evaluated using 1 fM to 1 µM of DNA linear targets (HPV-16-LT and HPV-18-LT). The specificity of the assay was evaluated using 1 pM of the DNA linear target. Complementary targets (HPV-16-LT and HPV-18-LT), mismatched targets (HPV-16-1m LT, HPV-16-3m LT, HPV-18-1m LT, HPV-18-3m LT), and non-complementary targets, which consisted of DNA sequences of other viruses such as Hepatitis A virus (HAV), Hepatitis B virus (HBV), Hepatitis C virus (HCV), and Hepatitis E virus (HEV).

### 2.9. Extraction of Genomic DNA from Clinical Samples

Forty oral samples consisting of 25 oral lichen planus (OLP), three oral leukoplakia (OL), 13 OSCC, and 14 normal mucosa and 17 cervical tissue samples were used in this study. Oral epithelial cells were collected using a soft toothbrush from the buccal mucosa. DNA extraction was performed using the Puregene^®^ Buccal Cell Core Kit A (Qiagen^®^, Mettmann, Germany) according to the manufacturer’s protocol. DNA from cervical brushes was extracted using the QIAamp DNA Mini Kit (Qiagen GmbH, Hilden, Germany) according to the manufacturer’s protocol. HeLa cell line containing the HPV18 and SiHa cell lines containing HPV-16 were used as HPV positive controls, while human white blood cells were used as the negative control. Plasmids containing cloned genome of HPV-16 and genomes of both HPV-16 and HPV-18 (referred to as mixed HPV-16/18) were also used as positive controls.

### 2.10. Nested PCR and Detection of HPV-16 and HPV-18 from Clinical Samples

HPV-16 and HPV-18 typing were performed using nested PCR [22] with the HPV-16 and HPV-18 outer and inner primer sets shown in Table 1. The nested PCR mixture contained 1× PCR Master Mix Solution, 0.3 µM of primers, and 50 ng of DNA template in a total volume of 25 μL reaction. lacking DNA template and non-HPV DNA (WBC) were used as negative controls, while plasmids containing the HPV-16 genome and mixed HPV-16/-18, and HeLa cell DNA were assigned as positive controls. First round of nested PCR was performed in a thermal cycler with the cycling parameters as follows: For HPV-16: 94 °C for 4 min, followed by 40 cycles (1 min at 94 °C, 1 min at 40 °C, and 1 min at 72 °C), with a final extension at 72 °C for 10 min; For HPV-18: 95 °C for 4 min, followed by 35 cycles (30 s at 95 °C, 30 s at 61 °C and 30 s at 72 °C), with a final extension at 72 °C for 1 min.

One µL of the first round (outer) PCR product was used as a template for second round (inner) PCR. The thermal cycling conditions for second round PCR were as follows: For HPV-16: 94 °C for 4 min, followed by 35 cycles (30 s at 94 °C, 30 s at 56 °C and 45 s at 72 °C), with a final extension at 72 °C for 4 min; For HPV-18: 95 °C for 4 min, followed by 35 cycles (30 s at 95 °C, 30 s at 63 °C and 30 s at 72 °C), with a final extension at 72 °C for 1 min. The PCR product was detected using an electrochemical biosensor and the result was compared with agarose gel electrophoresis. Briefly, 18 μL of 50 ng nested PCR product was mixed with 2 µL of CP-MNB. Hybridization and electrochemical signal detection were performed as previously described.

### 2.11. Ethical Consideration

The study involving the oral samples was reviewed and approved by the Ethics Committees of the Faculty of Dentistry and Faculty of Pharmacology, Mahidol University, Thailand (COA.No.MU-DT/PY-IRB 2020/008.1701 on 17 January 2020), while the study involving the use of cervical samples was reviewed and approved by the Ethics Committees of Khon Kaen University (HE562296) and Ubonratchathani Cancer Hospital, Thailand (EC012/2013). The institutional review board of the ethics committee for human research waived the need for consent because all samples were anonymous.

### 2.12. Statistical Analysis

Statistical analysis was performed using SPSS version 18.0. All electrochemical measurements were performed on five independent SPCEs, and the current responses are presented as mean values ± standard deviation (SD). The ANOVA test was used for the analysis, with the *p*-value of less than 0.05 considered statistically significant.

## 3. Results and Discussion

### 3.1. Characterization of the Silica Nanoparticles (SiNPs) and Silica Redox-Active Dye (Si-Dye)

Silica particles are widely used in electrochemical detection as carriers for the redox dye due to its stability, high loading capacity, ease of preparation, and ease of functionalizing the surface of the particles [20]. One study found that more MB molecules could be loaded per silica particle compared to polystyrene latex particles or multi-walled carbon nanotubes [21].

The redox-active dyes methylene blue (MB) and acridine orange (AO) were loaded into the silica nanoparticles through surface-reactive functional groups. The dyes interacted with the silica nanoparticles via strong electrostatic attractions between the positively charged dyes and the negatively charged silica. Furthermore, the dyes were also physically entrapped within the silica channels [20,21].

Next, a layer-by-layer procedure was used to create two oppositely charged bilayers on the Si-dye, starting with the deposition of positive-charged PAA followed by the negative-charged PSS (Figure 1). This enables avidin, which has a strong positive charge, to form an ionic interaction with the PSS layer. The use of biotin-labelled reporter probe allows for the binding of the DNA reporter probe to the Si-dye nanostructure via non-covalent avidin–biotin interaction.

Figure 1 shows the size distribution, scanning electron microscopy (SEM) image, and differential pulse voltammetry (DPV) peaks of SiNPs (without dye), SiMB, and SiAO. The core nanoparticles, SiNPs, had a mean diameter of 466.31 + 7.62 nm. The size increased to 506.45 + 3.98 nm and 518.57 + 4.21 nm after deposition of the MB and AO redox dyes, respectively. This agrees with the observations of Cheeveewattanagul et al. [21], which obtained Si core, Si-MB, and Si-AO diameter sizes of 450 ± 1.61; 598 ± 14.4, and 575 ± 2.21 nm, respectively. The SEM analysis of the SiNPs (without dye), SiMB, and SiAO is shown in Appendix A.

The size of the Si nanoparticles depends on the amount of tetraethyl orthosilicate (TEOS), ammonia, and water content in the reaction mixture [20]. Typically, this method produces particles in the 50 nm–2 μm range [23]. From our results, the sizes of the Si-core and Si-dye particles produced ranged from 460 to 520 nm, which was within the expected range. The size increase in Si-dye compared to the Si-core was 8.5% (Si-MB) and 11% (Si-AO). This size increase was similar to the results of Rossi et al. [20], who observed a 10–14% increase in the size of the Si-dye compared to the Si-core particles.

DPV analysis showed sharp peaks at −0.3 V (SiMB) and 0.7 V (SiAO), which were consistent with other studies [24,25]. The current responses of both Si-dyes were similar at a 10 mg/mL concentration. The use of MB- and AO-doped silica nanoparticles for single and multiplex target detection in electrochemical biosensor assays has been previously described [24,25].

### 3.2. Optimization of the Assay Conditions

Sandwich hybridization parameters such as amount of capture probe-magnetic bead (CP-MNB), reporter probe-conjugated to Si-dye/avidin (Si-RP), and hybridization time were optimized using 1 pM of complementary LT of each genotype or non-complementary target (non-HPV viruses). The optimization was performed to maximize the differences between positive and negative signals and decrease nonspecific binding. Statistical analysis was used to determine the best assay conditions. Figure 2 shows the results of the optimization experiments. The amount of CP-MNB varied between 0.50–2.50 µL, while all the other parameters were kept constant. The highest signal with 1 pM LT was achieved with 2.0 µL of CP-MNB (Figure 2a). Next, the amount of Si-RP varied between 0.50 and 2.50 µL, with 2.0 µL of CP-MNB while keeping all the other parameters constant. The best sensor performance was achieved with 20 µL of Si-RP (Figure 2b). Finally, using 20 µL of Si-RP and 2.0 µL of CP-MNB, hybridization time was optimized from 5–40 min. We found that 30 min hybridization time yielded the highest signal (Figure 2c). The optimized parameters were consistent with the values obtained in previous studies [21,24,25] that used a similar hybridization strategy.

### 3.3. Analytical Performance of the Assay

The analytical performance of the biosensor was performed using the optimized experimental conditions. The specificity of the assay was evaluated using 1 pM HPV-16 complementary, one base and three bases mismatch and non-complementary LT (Hepatitis A virus, HAV; Hepatitis B virus, HBV; Hepatitis C virus, HCV; and Hepatitis E virus, HEV). A current signal that was equal or greater than +3 standard deviations (3 SD) above the mean of the blank (background) signal was considered as a positive result. All reactions that contain the complementary target yielded a positive signal, while the non-complementary and mismatch targets were all negative (Figure 3a). Moreover, the assay could discriminate between perfectly complementary targets and one or three bases mismatch targets, demonstrating the highly specific nature of the assay. Analysis showed that the signals from blank, non-complementary, and mismatch targets were significantly different from the complementary targets.

The analytical sensitivity was determined using varying concentrations of complementary LT, from 1 fM to 1 µM. The calibration curve for both HPV-16 and HPV-18 showed a linear relationship with a correlation coefficient of 0.9912 between 1 fM and 1 µM (Figure 3b). The limits of detection (LOD) of HPV-16 and HPV-18 were 22 fM and 20 fM, respectively. The corresponding voltammograms showed that the current response increased as the concentration of LT increased (Figure 3c,d).

It is interesting to note that the LOD for both genotypes were similar, even though two different redox indicators were used. This could be due to both MB and AO having similar electrochemical responses on SPCE. This similarity in detection limit was also seen in our previous study where we used Si-MB and Si-AO for the multiplex detection of two different genes [24]. The LOD of the biosensor was compared with previously published DNA biosensors for HPV-16 and HPV-18 (Table 2). Our biosensor showed comparable results in terms of the limit of detection when compared to similar studies that utilized nanomaterials with electrochemical detection for HPV-16/18 [17,26].

### 3.4. Detection of HPV-16 and HPV-18 in Two Different Types of Clinical Samples

Previous studies utilizing biosensors for HPV detection have mainly evaluated their performance on cervical cancer samples [14,15,16,17,22,28,32]. Here, we demonstrated the utility of the electrochemical biosensor for the detection of two HPV genotypes in two different clinical samples (i.e., oral and cervical cancer samples).

A total of 55 oral samples and 25 cervical cancer samples were used in this evaluation. The nested PCR amplicons were analyzed using the biosensor assay and the conventional agarose gel electrophoresis method. For electrochemical detection, a current signal that was equal or greater than +3 standard deviations (3 SD) above the mean of the blank (background) signal was considered as positive. Figure 4 shows the result of HPV-16 and HPV-18 detection in the clinical samples. Two of the 13 OSCC samples were positive for HPV-16 and HPV-18, respectively. All of the other oral samples were negative.

For the cervical cancer samples, 10 were positive for HPV-16 only, five were positive for HPV-18 only, and six were positive for both HPV-16 and HPV-18. Six cervical samples were found to harbor both types of HPV. Several studies have shown that more than one type of HPV could be detected in oral [18] and cervical [22,27] samples. The comparison between the biosensor and agarose gel electrophoresis results was 100% concordant. The results are summarized in Table 3.

In this study, we have shown that the biosensor assay could detect HPV-16 and HPV-18 in two different sample matrices, which are the oral and cervical samples. Furthermore, we also evaluated different types of oral samples such as oral leukoplakia and oral lichen planus that have the potential to transform into OSCC. Instead of relying on different tests for oral and cervical cancers, this could simplify the molecular-based detection process for both cancers. The gold standard for OSCC and cervical cancer diagnosis is biopsy and histopathology, which only detect cancer lesion, but not the cause. HPV associated cancer may need different treatments and have a different prognosis. Therefore, this platform can be adjunct to the gold standard method. The whole assay, starting from target hybridization to electrochemical measurement, could be completed in 90 min.

## 4. Conclusions

The electrochemical biosensor could detect both high-risk HPV-16 and HPV-18 genotypes with high specificity and sensitivity in the low femtomolar range. Furthermore, the biosensor could be used for multianalyte detection of these two genotypes in oral and cervical cancer samples. Therefore, the biosensor could be used as a rapid and easy screening tool for the accurate detection of these oncogenic HPV types, which will certainly help in the diagnosis, management, and epidemiological study of cancer.

## Data Availability

The datasets generated during and/or analyzed during the current study are available from the corresponding author on reasonable request.

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
