# Peer review of "A Multianalyte Electrochemical Genosensor for the Detection of High-Risk HPV Genotypes in Oral and Cervical Cancers"

_biosensors, 2022, doi:10.3390/bios12050290_

Round 1

Reviewer 2 Report

The authors describe an interesting approach to detect two high-risk HPV genotypes (HPV-16 and HPV-18) in clinical samples. The developed DNA biosensor could be used with both oral and cervical samples that have different matrices and interesting results are presented for HPV detection. The research is of high importance, especially because the diagnostic tools is still limited for oral cancer. The manuscript would be suitable for publication by considering the following comments:

- I recommend to add schematic drawing of the designed electrochemical sensor shows the layer-by-layer procedure.

- section 2.7: define the abbreviation PB and add the concentration of KCL.

- Figure 3 (a): bars for HPV18-1mLT and HPV18-3mLT are missing.

Reviewer 3 Report

For its publication I consider it very important to improve the quality of the article. How some points:

Authors should include a schematic to facilitate understanding of biosensor development.

Section 3.2 Optimization of the assay conditions should be explained more clearly, relating all the parameters that appear in the figure 2.

The authors should carry out additional experiments related to the characterization of the surface, such as XPS or a study showing the type of interaction between the particles and the dyes. As well as explaining the interaction that occurs with the DNA sequences with the nanostructured surface.

What advantages do Si particles offer? There should be additional controls.

The authors should indicate the stability and durability of the biosensor.

As well as, indicate if the measurements are made in the same or in different electrodes.

It is advisable to describe figure 4. Specifically, figure captions b and d are impossible to understand.

Round 2

Reviewer 1 Report

The manuscript has gone through an extensive correction by the authors. I think the message of the study is now clear to the potential readers. 

With that, I would like to thank the authors for their patience with the accurate corrections. 

Best of luck

Author Response

The authors would like to thank the reviewer for the useful comments and for the acceptance of the manuscript.

Reviewer 3 Report

I believe that the article could be published in Biosensors.

Author Response

The authors would like to thank the reviewer for the useful comments and the minor spelling had been carefully checked.